

# Linear stability analysis of plane beds under flows with suspended load

Koji Ohata[1], Hajime Naruse[1], and Norihiro Izumi[2]

[1]Division of Earth and Planetary Sciences, Graduate School of Science, Kyoto University, Japan
[2]Division of Field Engineering for the Environment, Faculty of Engineering, Hokkaido University, Japan

**Correspondence:** Koji Ohata (ohata.koji.24z@gmail.com)

**Abstract.** Plane beds develop under flows in fluvial and marine environments; they are recorded as parallel lamination in sandstone beds, such as those found in turbidites. However, whereas turbidites typically exhibit parallel lamination, they rarely feature dune-scale cross lamination. Although the reason for the scarcity of dune-scale cross-lamination in turbidites is still debated, the formation of dunes may be dampened by suspended load. Here, we perform, for the first time, linear stability
analysis to show that flows with suspended load facilitate the formation of plane beds. For a fine-grained bed, suspended load can promote the formation of plane beds and dampen the formation of dunes. These results of theoretical analysis were verified with observational data of plane beds under open-channel flows. Our theoretical analysis found that suspended load promotes the formation of plane beds, which suggests that the development of dunes under turbidity currents is suppressed by the presence of suspended load.

## 1 Introduction

The interactions between fluids and erodible surfaces generate small-scale topographic features called bedforms both on terrestrial surfaces (e.g., riverbeds, deserts, and deep-sea floors) and on extra-terrestrial surfaces (Bourke et al., 2010; Gao et al., 2015; Hage et al., 2018; Cisneros et al., 2020). Such bedforms are preserved in sedimentary rocks as sedimentary structures such as cross- and parallel lamination (Harms, 1979). The types of sedimentary structures observed vary among different
types of rocks. Turbidites typically exhibit parallel lamination (Bouma, 1962), whereas they rarely feature dune-scale cross-lamination (Talling et al., 2012). However, the opposite is true for fluvial deposits; i.e., dune-scale cross laminae are often observed in riverine sandstone (Miall, 2010).

Although the reason for the paucity of dune-scale cross-lamination in turbidites is still debated (Lowe, 1988; Arnott, 2012; Schindler et al., 2015; Tilston et al., 2015), it could be attributed to the presence of suspended load. For example, in the case
of open-channel flows, nearly flat bed waves and low-angle dunes have been observed in suspension-dominated rivers (Smith and McLean, 1977; Kostaschuk and Villard, 1996; Bradley et al., 2013; Ma et al., 2017). Additionally, flume experiments have suggested that dune height decreases with increasing suspended load flux (Bridge and Best, 1988; Naqshband et al., 2017). Therefore, the influence of suspended load on the suppression of dune development and the formation of plane beds is worth investigating.



The relationships between sediment transport modes and the formation of plane beds have received little attention in theoretical works that performed linear stability analyses. The reason could be because previous studies have succeeded in predicting the wavelength of dunes and antidunes without considering suspended load (Colombini, 2004; Di Cristo et al., 2006; Colombini and Stocchino, 2008; Vesipa et al., 2012; Bohorquez et al., 2019). However, this assumption is not appropriate for analyzing open-channel flows where the suspended load is not negligible, such as flows in rivers with a fine sediment bed (de Almeida

et al., 2016; Sambrook Smith et al., 2016). Moreover, although some research has considered both bed- and suspended load (Engelund, 1970; Nakasato and Izumi, 2008; Bose and Dey, 2009), the hydraulic conditions of these analyses were limited, and the results were tested using only observational data of dunes and antidunes.

  Therefore, in order to investigate the effect of sediment transport mode on the formation of plane beds, we performed a linear stability analysis of bedforms under open-channel flows carrying suspended load. The model introduced in Nakasato

and Izumi (2008) was extended in this study to evaluate plane bed formation under various conditions of sediment diameter and flow depth. To evaluate the suspended load effect, linear stability analyses were performed on flows both with and without suspended load. Further, we tested our stability diagrams against observational data of plane beds. Our theoretical analysis reveals for the first time that suspended load promotes the formation of plane beds, which has implications for interpreting sedimentary structures in turbidites.

## 2 Methods


  Linear stability analysis of fluvial bedforms can provide the wavelengths of perturbations (i.e., bed waves) that grow over time (Colombini, 2004; Bohorquez et al., 2019). We employ the two-dimensional Reynolds-averaged Navier-Stokes equations as the governing equations for flows and the quasi-steady assumption to neglect the unsteady terms in the flow equations. The eddy viscosity is evaluated using a mixing-length approach. In this study, bed-load discharge is estimated using the Meyer-Peter

and Müller formula modified as described in Wong and Parker (2006). The entrainment rate of suspended load is estimated using the relationship proposed in de Leeuw et al. (2020). See the following section for details. To test the results of linear stability analyses against the observational data of plane beds, we plotted stability diagrams in the parametric space of hydraulic parameters.

### 2.1 Governing Parameters

The instability of a system is illustrated as a contour diagram of the perturbation growth rate $\omega_i$ (Fig. 1). Generally, theoretical studies of bedforms based on linear stability analyses describe the transition of bedform phases in the parametric space of wavenumber $k$ and Froude number Fr, which are given by:

$$k = \frac{2\pi \tilde{h}_0}{\tilde{\lambda}} \tag{1}$$

$$\mathrm{Fr} = \frac{\tilde{U}_0}{\sqrt{\tilde{g}\tilde{h}_0}} \tag{2}$$





where $\tilde{\lambda}$ denotes the perturbation wavelength, $\tilde{U}_0$ is the depth-averaged flow velocity of the uniform flow, $\tilde{g}$ is the gravitational acceleration ($= 9.81 \text{ m}^2/\text{s}$), and $\tilde{h}_0$ is the flow depth of the uniform flow. Hereafter, we denote dimensional variables using a tilde ($\tilde{\ }$).

Stability diagrams described on the $k$-Fr plane have been commonly used to predict the development of dunes and antidunes (Kennedy, 1963). A few studies have used other combinations of dimensionless numbers such as the friction coefficient $C$
versus Fr (Colombini and Stocchino, 2008) and the relative roughness $\tilde{D}/\tilde{h}_0$ on the $k$-Fr plane (Bohorquez et al., 2019).

Although the classic $k$-Fr diagrams are widely accepted, we cannot use this approach to evaluate whether plane bed formation can be predicted reliably because plane beds have extremely small wavenumber or have an infinite wavelength (i.e., they are flat). Therefore, we illustrate stability diagrams as contour maps of dominant wavenumber $k_\mathrm{d}$ on the $\tilde{D}/\tilde{h}_0$-Fr plane with fixed $\tilde{D}$ and the $\mathrm{Re_p}$-Fr plane with fixed $\tilde{h}_0$ to investigate the impact of suspended load on the formation of plane beds, where
the dominant wavenumber $k_\mathrm{d}$ denotes the wavenumber that provides maximum growth rate.

The instability of a system is illustrated as a contour diagram of the perturbation growth rate $\omega_\mathrm{i}$ (Fig. 1). We can rewrite Eq. (A30) as:

$$\omega = \omega\left(k, \mathrm{Fr}, \tilde{D}, \tilde{h}_0\right) \tag{3}$$

Thus, we can obtain the growth rate $\omega_\mathrm{i}$ as a function of $k$ for a given combination of $\left(\mathrm{Fr}, \tilde{D}, \tilde{h}_0\right)$. In this study, we assume
that the system is stable if $\omega_\mathrm{i}$ is not positive for all $k$ within the domain $[k_\mathrm{min}, k_\mathrm{max}]$ for a given $\left(\mathrm{Fr}, \tilde{D}, \tilde{h}_0\right)$ combination. In contrast, the system is assumed to be unstable if $\omega_\mathrm{i}$ is positive for some $k$ (Fig. 1). We describe stability diagrams as contour maps of $k_\mathrm{d}$ in the parametric space of $\left(\mathrm{Fr}, \tilde{D}/\tilde{h}_0\right)$ (Figs. 2 and 3) and $(\mathrm{Fr}, \mathrm{Re_p})$ (Figs. 4 and 5).

Therefore, we employed (1) two grades of fine particles ($\tilde{D} = 0.12$ and $0.25$ mm) and one grade of coarse particles ($\tilde{D} = 1.2$ mm) and (2) two grades of shallow flow depth ($\tilde{h}_0 = 0.15$ and $0.30$ m) and one grade of deep flow depth ($\tilde{h}_0 = 1.0$ m) to
investigate the effect of suspension on the bed instability. The Froude number, particle diameter, and flow depth range from 0.01 to 2, 0.125 mm to 4 mm, and 1 cm to 5.0 m, respectively. The domain $[k_\mathrm{min}, k_\mathrm{max}]$ was set as $[0.01, 1.5]$, corresponding to $\lambda$ ranging from $\sim 4.2h$ to $\sim 628h$.

## 2.2  Linear Stability Analysis

Here we present the formulation of the problem and the method used to solve the differential equations.

### 2.2.1  Flow equations

The governing equations for flows are the two-dimensional Reynolds-averaged Navier-Stokes equations. On erodible beds, the flow adjustments occur immediately relative to the bed adjustments (Fourrière et al., 2010). Therefore, we employ the quasi-steady assumption to neglect the unsteady terms in the flow equations (Colombini, 2004; Yokokawa et al., 2016).



Under the quasi-steady assumption, the dimensionless forms of the Reynolds-averaged Navier-Stokes equations and continuity equation for incompressible flow are described as:

$$u\frac{\partial u}{\partial x} + w\frac{\partial u}{\partial z} = -\frac{\partial p}{\partial x} + 1 + \frac{\partial T_{xx}}{\partial x} + \frac{\partial T_{xz}}{\partial z} \tag{4}$$

$$u\frac{\partial w}{\partial x} + w\frac{\partial w}{\partial z} = -\frac{\partial p}{\partial z} + S^{-1} + \frac{\partial T_{xz}}{\partial x} + \frac{\partial T_{zz}}{\partial z} \tag{5}$$

$$\frac{\partial u}{\partial x} + \frac{\partial w}{\partial z} = 0 \tag{6}$$

where $u$ and $w$ are the flow velocities in $x$- and $z$- direction, respectively; $p$ denotes the pressure; $S$ is the bed slope; and $T_{ij}$ $(i,j = x,z)$ is the Reynolds stress tensor.

We employ a Boussinesq-type assumption to close the flow equations:

$$T_{xx} = 2\nu_T \frac{\partial u}{\partial x} \tag{7}$$

$$T_{zz} = 2\nu_T \frac{\partial w}{\partial z} \tag{8}$$

$$T_{xz} = \nu_T \left(\frac{\partial u}{\partial x} + \frac{\partial w}{\partial z}\right) \tag{9}$$

Then, the eddy viscosity $\nu_T$ is evaluated using a mixing-length approach:

$$\nu_T = l^2 \left|\frac{\partial u}{\partial z}\right| \tag{10}$$

$$l = \kappa(z - Z)\sqrt{\frac{h + R - z}{h}} \tag{11}$$

where $l$ is the mixing length, $\kappa$ is the Kármán coefficient (= 0.4), $h$ is the flow depth, $Z$ denotes the bed height, and $R$ is the height of the reference level at which the flow velocity is assumed to vanish in a logarithmic profile (Fig. A1).

In the above equations, the system is nondimensionalized as follows:

$$(u, w) = (\tilde{u}, \tilde{w})/\tilde{u}_{f0} \tag{12}$$

$$(x, z, h, Z, R, D) = (\tilde{x}, \tilde{z}, \tilde{h}, \tilde{Z}, \tilde{R}, \tilde{D})/\tilde{h}_0 \tag{13}$$

$$(p, T_{ij}) = (\tilde{p}, \tilde{T}_{ij})/\tilde{\rho}\tilde{h}_0 \tag{14}$$

$$\nu_T = \tilde{\nu}_T/(\tilde{u}_{f0}\tilde{h}_0) \tag{15}$$

where $D$ is the non-dimensional diameter of a bed particle, $\tilde{u}_{f0}$ denotes the shear velocity in the basic flat-bed state, and $\tilde{\rho}$ is the water density (= 1000 kg/m³). The shear velocity in the basic flat-bed state $\tilde{u}_{f0}$ is obtained as:

$$\tilde{u}_{f0} = \sqrt{\tilde{g}\tilde{h}_0 S} \tag{16}$$

As the flow is continuous, the system can be rewritten using the stream function $\psi$ defined as:

$$(u, w) = \left(\frac{\partial \psi}{\partial z}, -\frac{\partial \psi}{\partial x}\right) \tag{17}$$





Then, Eqs. (4) and (5) are rearranged to:

$$
\frac{\partial \psi}{\partial z}\frac{\partial^2 \psi}{\partial x \partial z} - \frac{\partial \psi}{\partial x}\frac{\partial^2 \psi}{\partial z^2} = -\frac{\partial p}{\partial x} + 1 + \frac{\partial}{\partial x}\left(2\nu_T \frac{\partial^2 \psi}{\partial x \partial z}\right)
$$
$$
+ \frac{\partial}{\partial z}\left[\nu_T\left(\frac{\partial^2 \psi}{\partial z^2} - \frac{\partial^2 \psi}{\partial x^2}\right)\right]
$$
(18)

$$
\frac{\partial \psi}{\partial x}\frac{\partial^2 \psi}{\partial x \partial z} - \frac{\partial \psi}{\partial z}\frac{\partial^2 \psi}{\partial x^2} = -\frac{\partial p}{\partial z} + S^{-1} - \frac{\partial}{\partial z}\left(2\nu_T \frac{\partial^2 \psi}{\partial x \partial z}\right)
$$
$$
+ \frac{\partial}{\partial x}\left[\nu_T\left(\frac{\partial^2 \psi}{\partial z^2} - \frac{\partial^2 \psi}{\partial x^2}\right)\right]
$$
(19)

Eliminating $p$ from Eqs. (18) and (19), we obtain:

$$
\frac{\partial \psi}{\partial z}\frac{\partial}{\partial x}\nabla^2 \psi - \frac{\partial \psi}{\partial x}\frac{\partial}{\partial z}\nabla^2 \psi - 4\frac{\partial^2}{\partial x \partial z}\left(\nu_T \frac{\partial^2 \psi}{\partial x \partial z}\right)
$$
$$
+ \left(\frac{\partial^2}{\partial x^2} - \frac{\partial^2}{\partial z^2}\right)\left[\nu_T\left(\frac{\partial^2}{\partial z^2} - \frac{\partial^2}{\partial x^2}\right)\psi\right] = 0
$$
(20)

### 2.2.2 Advection-diffusion equations for suspended sediment

We also assume a quasi-steady state for the advection-diffusion equation for suspended sediment, which is formulated as:

$$
\frac{\partial F_x}{\partial x} + \frac{\partial F_z}{\partial z} = 0
$$
(21)

Here, $F_x$ and $F_z$ are the normalized fluxes of suspended sediment in $x$- and $z$- directions, respectively, given by:

$$
F_x = uc - \nu_T \frac{\partial c}{\partial x}
$$
(22)

$$
F_z = (w - w_{\mathrm{s}})c - \nu_T \frac{\partial c}{\partial z}
$$
(23)

where $c$ denotes the concentration of suspended sediment and $w_{\mathrm{s}}$ is the settling velocity of sediment. We assume that the diffusion coefficient of suspended sediment is equal to the eddy viscosity $\nu_T$. Based on Eqs. (22) and (23), Eq. (21) is reformulated as:

$$
u\frac{\partial c}{\partial x} + (w - w_{\mathrm{s}})\frac{\partial c}{\partial z} = \frac{\partial}{\partial x}\left(\nu_T \frac{\partial c}{\partial x}\right) + \frac{\partial}{\partial z}\left(\nu_T \frac{\partial c}{\partial z}\right)
$$
(24)

The settling velocity of sediment $w_{\mathrm{s}}$ is calculated using a relationship given in Ferguson and Church (2004):

$$
w_{\mathrm{s}} = \frac{\tilde{w}_{\mathrm{s}}}{\sqrt{R_{\mathrm{s}}\tilde{g}\tilde{D}}}
$$
(25)

$$
\tilde{w}_{\mathrm{s}} = \frac{R_{\mathrm{s}}\tilde{g}\tilde{D}^2}{C_1\tilde{\nu} + 0.75C_2\sqrt{R_{\mathrm{s}}\tilde{g}\tilde{D}^3}}
$$
(26)

where the constants $C_1$ and $C_2$ are set to the values for natural sand: $C_1 = 18$ and $C_2 = 1.0$.



Earth **Surface**
**Dynamics**
Discussions



The particle Reynolds number $\mathrm{Re_p}$ is defined as:

$$\mathrm{Re_p} = \frac{\sqrt{R_\mathrm{s}\tilde{g}\tilde{D}^3}}{\tilde{\nu}} \tag{27}$$

where $R_\mathrm{s}$ is the submerged specific density and $\tilde{\nu}$ is the kinematic viscosity of the fluid ($= 1.0 \times 10^{-6}\ \mathrm{m^2/s}$). The submerged

specific density $R_\mathrm{s}$ is defined as:

$$R_\mathrm{s} = \frac{\tilde{\rho}_\mathrm{s} - \tilde{\rho}}{\tilde{\rho}} \tag{28}$$

where $\tilde{\rho}_\mathrm{s}$ denotes the density of the bed particles ($= 2650\ \mathrm{kg/m^3}$).

### 2.2.3  Transformation of variables

We employ the following transformation of variables to apply the boundary condition at the bed and flow surfaces:

$$\xi = x \tag{29}$$

$$\eta = \frac{z - R(x)}{h(x)} \tag{30}$$

The derivatives with respect to $x$ and $z$ are described as follows:

$$\frac{\partial}{\partial x} = \frac{\partial}{\partial \xi} - \frac{\eta \partial_x h + \partial_x R}{h}\frac{\partial}{\partial \eta} \tag{31}$$

$$\frac{\partial}{\partial z} = \frac{1}{h}\frac{\partial}{\partial \eta} \tag{32}$$

where $\partial_x$ denotes the partial derivative with respect to $x$. Using the above transformation of variables approach, the height of
the water surface and the reference level correspond to $\eta = 1$ and $\eta = 0$, respectively.

Additionally, the dimensionless mixing length $l$ (Eq. (11)) is rearranged as:

$$l = \kappa(h\eta + R - Z)\sqrt{\frac{1 - \eta}{1 + (R - Z)/h}} \tag{33}$$

Since $(R - Z)/h \ll 1$, then we can obtain:

$$l = \kappa(h\eta + R - Z)\sqrt{1 - \eta} \tag{34}$$

### 2.2.4  Boundary condition

The boundary conditions include a vanishing flow component normal to the water surface, and vanishing stresses normal and
tangential to the water surface as follows:

$$\left.\begin{aligned} \boldsymbol{u} \cdot \boldsymbol{e}_\mathrm{ns} &= 0 \\ \boldsymbol{e}_\mathrm{ns} \cdot \mathbf{T} \cdot \boldsymbol{e}_\mathrm{ns} &= 0 \\ \boldsymbol{e}_\mathrm{ts} \cdot \mathbf{T} \cdot \boldsymbol{e}_\mathrm{ns} &= 0 \end{aligned}\right\} \quad \text{at} \quad \eta = 1 \tag{35}$$





where $\boldsymbol{u} = (u, w)$ is the velocity vector, $\boldsymbol{e}$ denotes the unit vector, and $\mathbf{T}$ is the stress tensor. The subscripts ns and ts denote directions normal and tangential to the water surface, respectively.

At the bed, the boundary conditions include the vanishing flow components normal and tangential to the bed.

$$\left.\begin{array}{l} \boldsymbol{u} \cdot \boldsymbol{e}_{\text{nb}} = 0 \\ \boldsymbol{u} \cdot \boldsymbol{e}_{\text{tb}} = 0 \end{array}\right\} \quad \text{at} \quad \eta = 0 \tag{36}$$

where the subscripts nb and tb denote directions normal and tangential to the bed, respectively. The vectors $\boldsymbol{e}_{\text{ns}}$, $\boldsymbol{e}_{\text{ts}}$, $\boldsymbol{e}_{\text{nb}}$, and $\boldsymbol{e}_{\text{tb}}$, and the tensor $\mathbf{T}$ are defined as:

$$\boldsymbol{e}_{\text{ns}} = \frac{1}{\sqrt{1 + \partial_x (R+h)^2}} \left( -\partial_x (R+h), 1 \right) \tag{37}$$

$$\boldsymbol{e}_{\text{ts}} = \frac{1}{\sqrt{1 + \partial_x (R+h)^2}} \left( 1, \partial_x (R+h) \right) \tag{38}$$

$$\boldsymbol{e}_{\text{nb}} = \frac{1}{\sqrt{1 + \partial_x R^2}} \left( -\partial_x R, 1 \right) \tag{39}$$

$$\boldsymbol{e}_{\text{tb}} = \frac{1}{\sqrt{1 + \partial_x R^2}} \left( 1, \partial_x R \right) \tag{40}$$

$$\mathbf{T} = \begin{pmatrix} -p + T_{xx} & T_{xz} \\ T_{xz} & -p + T_{zz} \end{pmatrix} \tag{41}$$

The boundary conditions for the suspended sediment flux at the flow surface and bed are as follows:

$$\boldsymbol{F} \cdot \boldsymbol{e}_{\text{ns}} = 0 \quad \text{at} \quad \eta = 1 \tag{42}$$

$$\boldsymbol{F} \cdot \boldsymbol{e}_{\text{nb}} = \frac{\tilde{E}_{\text{s}}}{\tilde{u}_{\text{f0}}} \quad \text{at} \quad \eta = 0 \tag{43}$$

where $\boldsymbol{F} = (F_x, F_z)$ is the flux vector of suspended sediment and $\tilde{E}_{\text{s}}$ is the entrainment rate of the sediment calculated as $\tilde{E}_{\text{s}} = \tilde{w}_{\text{s}} E_{\text{s}}$. In this study, the dimensionless coefficient $E_{\text{s}}$ is estimated using the relationship proposed in de Leeuw et al. (2020):

$$E_{\text{s}} = C_3 \left( \frac{u_{\text{f}}}{w_{\text{s}}} \right)^{e_1} \text{Fr}^{e_2} \text{Re}_{\text{p}}^{e_3} \tag{44}$$

where $C_3$ was set to $5.73 \times 10^{-3}$ and coefficients $e_1$, $e_2$, and $e_3$ were set to $1.31$, $1.59$, and $-0.86$, respectively.

### 2.2.5 Basic state

The basic flow state for linear stability analysis is a uniform flow over a flat bed. Under this condition, the hydraulic parameters $u$, $w$, $h$, $Z$, $R$, and $c$ are described as:

$$(u, w, h, Z, R, c) = (u_0(\eta), 0, 1, 0, R_0, c_0(\eta)) \tag{45}$$





where the subscript 0 denotes a parameter in the basic state. The governing equations of flows can be simplified as:

$$1 + \frac{\partial T_{xy0}}{\partial \eta} = 0 \tag{46}$$


$$T_{xy0} = \nu_{T0} \frac{\partial u_0}{\partial \eta} \tag{47}$$

$$\nu_{T0} = l_0^2 \frac{\partial u_0}{\partial \eta} \tag{48}$$

$$l_0 = \kappa(\eta + R_0)\sqrt{1-\eta} \tag{49}$$

with the boundary conditions:

$$u_0 = 0, \ T_{xy0} = 1 \quad \text{at} \quad \eta = 0 \tag{50}$$

With Eqs. (46)–(50), we can obtain the following logarithmic law for the flow velocity:

$$u_0(\eta) = \frac{1}{\kappa} \ln\left(\frac{\eta + R_0}{R_0}\right) \tag{51}$$

Then, the friction coefficient $C_z$ is obtained by the direct integration of Eq. (51) from $\eta = 0$ to $\eta = 1$:

$$C_z = \frac{\tilde{U}_0}{\tilde{u}_{f0}} = \frac{1}{\kappa}\left[(1+R_0)\ln\left(\frac{1+R_0}{R_0}\right) - 1\right] \tag{52}$$

Now, we consider the logarithmic law of the open-channel flows as:


$$u = \frac{1}{\kappa}\ln\left(\frac{z}{z_0}\right) \tag{53}$$

with $z_0 = D/12$ (Colombini, 2004). It should be noted here that the bed roughness can be modified by the sediment transport (Dietrich and Whiting, 1989). Additionally, we set the origin of $z$-axis at a distance of $D/6$ below the top of the bed particles (Fig. A2). By setting the top of the bed particles as $z = D/6$, the reference level $R_0$ is positioned below the top of bed particles. Therefore, the domain in which the mixing-length approach cannot be applied is restricted near the bed.

Under the above uniform flow condition over a flat bed, Eq. (24) can be rewritten as:

$$-w_s\frac{\partial c_0}{\partial \eta} = \frac{\partial}{\partial \eta}\left(\nu_{T0}\frac{\partial c_0}{\partial \eta}\right) \tag{54}$$

with the following boundary conditions:

$$w_s c_0 + \nu_{T0}\frac{\partial c_0}{\partial \eta} = 0 \quad \text{at} \quad \eta = 1 \tag{55}$$

$$c_0 = c_b \quad \text{at} \quad \eta = 0 \tag{56}$$

Here, $c_b$ is the near-bed concentration of suspended sediment. Under the basic state, the entrainment and deposition rates of the suspended sediment are balanced. Thus, $c_b$ is described as:

$$c_b = E_{s0} \tag{57}$$

$$E_{s0} = C_3\left(\frac{u_{f0}}{w_s}\right)^{e_1} \text{Fr}^{e_2} \text{Re}_p^{e_3} \tag{58}$$





where $C_3$ was set to $5.73 \times 10^{-3}$ and coefficients $e_1$, $e_2$, and $e_3$ were set to 1.31, 1.59, and $-0.86$, respectively.

By integrating Eq. (54), we obtain the suspended sediment distribution in the basic state as follows:

$$c_0(\eta) = c_{\mathrm{b}} \left[ \frac{R_0(1-\eta)}{\eta + R_0} \right]^{w_{\mathrm{s}}/\kappa(1+R_0)} \tag{59}$$

### 2.2.6   Temporal development of bed configurations

The development of the bed configuration can be described by the Exner equation considering the suspended load as follows:

$$(1-\lambda_{\mathrm{p}})\frac{\partial \tilde{B}}{\partial \tilde{t}} + \alpha_{\mathrm{b}}\frac{\partial \tilde{q}_{\mathrm{B}}}{\partial \tilde{x}} + \alpha_{\mathrm{s}}\tilde{w}_{\mathrm{s}}\left(E_{\mathrm{s}} - c_{[\xi,\eta_b]}\right) = 0 \tag{60}$$

where $\lambda_p$ denotes the sediment porosity, $\tilde{B}$ denotes the height of the bed-load layer, $\tilde{t}$ is time, and $\tilde{q}_B$ denotes the bed-load discharge per unit width. In the case without suspension, the development of the bed configuration associated with suspended load is ignored by setting the coefficient $\alpha_{\mathrm{s}}$ in Eq. (60) to 0. In the case of the stability analysis with suspension, the coefficient $\alpha_{\mathrm{s}}$ take a value of 0 or 1 depending on the sediment transport regime (Eq. (71)).

Equation (60) is nondimensionalized as:

$$\frac{\partial B}{\partial t} + \alpha_{\mathrm{b}}\frac{\partial q_{\mathrm{B}}}{\partial \xi} + \alpha_{\mathrm{s}}\frac{w_{\mathrm{s}}}{D}\left(E_{\mathrm{s}} - c_{[\xi,\eta_b]}\right) = 0 \tag{61}$$

with

$$\tilde{t} = \frac{(1-\lambda_{\mathrm{p}})\tilde{h}_0^{\,2}}{\sqrt{R_{\mathrm{s}}\tilde{g}\tilde{D}^3}}t \tag{62}$$

In this study, dimensionless bed-load discharge per unit width is estimated using the Meyer-Peter and Müller formula modified as described in Wong and Parker (2006); this equation is given as:

$$q_{\mathrm{B}} = \frac{\tilde{q}_{\mathrm{B}}}{\sqrt{R_{\mathrm{s}}g\tilde{D}^3}} = C_4(\theta_{\mathrm{b}} - \theta_{\mathrm{c}})^{e_4} \tag{63}$$

where $C_4$ and $e_4$ were set to 3.97 and 1.5, respectively. Here, $\theta_{\mathrm{b}}$ is the Shields stress at the top of bed-load layer and $\theta_{\mathrm{c}}$ is the critical Shields stress for particle motion. These variables can be expressed as follows:

$$\theta_0 = \frac{S}{R_{\mathrm{s}}D} \tag{64}$$

$$\theta_{\mathrm{b}} = \theta_0 \tau_{\mathrm{b}} \tag{65}$$

$$\theta_{\mathrm{c}} = \theta_{\mathrm{ch}} - \mu\left(S - \frac{\partial B}{\partial x}\right) \tag{66}$$

where $\theta_0$ is the Shields stress of the base flow, $\tau_{\mathrm{b}}$ denotes the shear stress at the top of the bed-load layer, $\theta_{\mathrm{ch}}$ denotes the critical Shields stress under the flat-bed conditions, and $\mu$ is a constant set to 0.1 (Fredsøe, 1974). The shear stress $\tau_{\mathrm{b}}$ is described as:

$$\tau_{\mathrm{b}} = \left[e_{\mathrm{tb}} \cdot \mathbf{T} \cdot e_{\mathrm{nb}}\right]_{\eta=\eta_{\mathrm{b}}} \tag{67}$$





where $\eta_b$ is the dimensionless thickness of the bed-load layer and is obtained as:

$$\eta_{\mathrm{b}} = B_0 - R_0 = h_{\mathrm{b}} + \frac{D}{12} \tag{68}$$

where $B_0$ and $R_0$ denote the height of the top of the bed-load layer and the reference level in the basic state, respectively. According to Colombini (2004), the thickness of the bed-load layer $h_{\mathrm{b}}$ is estimated as follows:

$$h_{\mathrm{b}} = l_{\mathrm{b}} D \tag{69}$$

$$l_{\mathrm{b}} = 1 + 1.3 \left( \frac{\tau_{\mathrm{r}} - \tau_{\mathrm{c}}}{\tau_c} \right)^{0.55} \tag{70}$$

where $l_{\mathrm{b}}$ denotes the relative saltation height, $\tau_{\mathrm{r}}$ is the shear stress at the reference level, and $\tau_{\mathrm{c}}$ is the critical shear stress.

In this study, the sediment transport regimes are classified using the threshold conditions of sediment motion in Brownlie (1981) as follows:

$$\theta_{\mathrm{ch}} = 0.22 \mathrm{Re_p}^{-0.6} + 0.06 \exp(-17.77 \mathrm{Re_p})^{-0.6} \tag{71}$$

The coefficients $\alpha_{\mathrm{b}}$ and $\alpha_{\mathrm{s}}$ in Eq. (60) were set to 0 when $\theta_0 < \theta_{\mathrm{ch}}$ and set to 1 when $\theta_{\mathrm{ch}} \leq \theta_0$.

## 2.2.7 Linear Analysis

We impose an infinitesimal perturbation on the basic state. Then, with the use of boundary conditions, we can solve the differential equations to get the growth rate of the perturbation. Please see the appendix for details of linear analysis.

## 2.3 Compilation of published data

The stability diagrams were assessed using an observational dataset pertaining to open-channel flows compiled from the literature, as summarized in Tables A1–A6. We compiled from the literature a total of 269 sets of data for Figures 2 and 3 and 276 sets of data for Figures 4 and 5. The flow depth, the flow velocity, and particle diameter ranges from 0.02 to 19.5 m, 0.198 to 1.99 m/s, and 0.096 to 1.6 mm, respectively.

We used the data of plane beds in which the sediment transport mode could be identified, i.e., plane bed without suspension, with suspension, and with sheet flows. We identified whether sediment particles were transported as suspended load or not based on the suspended sediment concentration. Plane bed without sediment movement were not included in this analysis. For comparison with the theoretical analysis results, we used the data of dunes and antidunes with wavenumbers with the range $0 < k \leq 1.5$ for comparison.

For Figures 2 and 3, the data of which sediment diameter range from $0.74\tilde{D}$ to $1.36\tilde{D}$ were chosen to plot on stability diagram, which corresponds to the range $\log_{10} \mathrm{Re_p} \pm 0.2$. The data of which flow depth range from $0.71\tilde{h}_0$ to $1.41\tilde{h}_0$ were chosen to plot Figures 4 and 5. To calculate the particle Reynolds number, the kinematic viscosity $\nu$ was assumed as follows (van den Berg and van Gelder, 1993):

$$\nu = \left[ 1.14 - 0.031 \left( T - 15 \right) + 0.00068 \left( T - 15 \right)^2 \right] 10^{-6} \tag{72}$$

where $T$ represents the water temperature in degrees Celsius. A value of $20°$C was assumed for data when $T$ was not reported.





## 3   Results

### 3.1   $\tilde{D}/\tilde{h}_0$-Fr diagram

The contour maps of $k_\mathrm{d}$ on $\tilde{D}/\tilde{h}_0$-Fr plane show that the stable region, which denotes that hydraulic conditions where the plane bed appear, for fine sediments is larger in the diagram with suspension than in that without suspension (Fig. 2). A stable region appears at $0.6 < \mathrm{Fr} < 1.2$ and for $h < 1.2$ m in the case of the stability analysis without suspension (Fig. 2a, c), and the dominant wavenumber increases with increasing flow depth. In the case of the stability analysis with suspension, Froude number and flow depth of the stable region ranges from 0.05 to 1.0 and from 0.01 to 5, respectively (Fig. 2b, d), while at $\mathrm{Fr} > 1$, the dominant wavenumber can fall below 0.3 (Fig. 2b, d).

Comparing the results where $\tilde{D} = 0.12$ mm and the observational data, in the case without suspension, all the plane bed data are within unstable region; most values plot in the region where $k_\mathrm{d} > 1$ (Fig. 2a). In contrast, all the plane bed data plot in the stable region in the case with suspension (Fig. 2b). When $\tilde{D} = 0.25$ mm, the plane bed data without suspension plot below the threshold of sediment motion (Fig. 2c, d). Moreover, although some observational data points of plane beds with suspension plot within the stable region in both diagrams, more data agree with the stable region in the case with suspension than in that without suspension (Fig. 2c, d). As expected, most dune and antidune data plot in the unstable region, whereas several data points of dunes and antidunes plot in the stable region in both cases with and without suspension (Fig. 2).

The stability diagrams considering flows with and without suspension for coarse sediment beds ($\tilde{D} = 1.20$ mm) differ in the region where $\mathrm{Fr} < 1$ (Fig. 3). Contrary to the case of fine sand, the unstable region is wider in the diagram with suspension than in that without suspension. The data of plane beds without suspended load plot just above the threshold of sediment motion (Fig. 3). The observational data of plane beds under sheet flows fall inside the unstable region where $0.3 < k_\mathrm{d} < 0.5$ and $\mathrm{Fr} > 1.6$ (Fig. 3).

### 3.2   $\mathrm{Re_p}$-Fr diagram

The contour maps of $k_\mathrm{d}$ on $\mathrm{Re_p}$-Fr plane show that the stable region is larger at $\mathrm{Re_p} < 20$ in the diagram with suspension than in that without suspension (Figs. 4 and 5). Comparing with the observational data, most plane bed data plot in the stable region in the case of the stability analysis with suspension (Figs. 4b, d and 5b), although some observational data points of plane beds with suspension plot within the unstable region. Figures 4 and 5 also show that most dune and antidune data plot in the unstable region.

## 4   Discussion

### 4.1   Effect of suspension on fine sediment bed

The role of suspended load in the formation of plane beds and suppressing dune-scale instabilities is quantitatively illustrated as the broadening of the stable regions (Figs. 2, 4 and 5). The stability diagrams for fine sediment beds show a good agreement with





the observational data of plane beds under flows with suspension (Figs. 2b, d, 4b, d and 5b). The transition from dunes to plane

beds has been explained by the spatial lag $\delta$ between the bed topography and the local sediment transport rate (Naqshband et al., 2014; van Duin et al., 2017). If the bed topography and sediment transport rate are entirely in-phase ($\delta = 0$), dunes migrate downstream without growth or decay. The dune height increases and decreases when the maximum sediment transport rate occurs upstream ($\delta < 0$) and downstream ($\delta > 0$) of the dune crest, respectively. Kennedy (1963) introduced the spatial lag in his flow model to account for the bedform growth and decay, and subsequent research has investigated the effect of spatial lag

on the bedform development (McLean, 1990; van Duin et al., 2017). Recently, Naqshband et al. (2017) quantitatively observed the positive spatial lag under suspended load dominated flows in their flume experiments. Our analyses confirm that suspended load dampens the development of bed waves, thereby facilitating the formation of plane beds, and thus cannot be neglected in theoretical analyses for realistic predictions of bedforms.

We found that dunes are deformed under flows with suspended load, although further work is needed to investigate the

amplitudes of dunes under such conditions. Field surveys have indicated the existence of low-angle dunes in suspended-load dominated rivers (Smith and McLean, 1977; Kostaschuk and Villard, 1996; Hendershot et al., 2016); moreover, flume experiments have indicated that dune height decreases with increasing suspended load flux (Naqshband et al., 2017; Bradley and Venditti, 2019). Theoretical analyses in Fredsøe (1981) have also predicted a decrease of dune steepness under unsteady flows with suspension where the flow discharges were being increased. In future works, nonlinear analyses should be done to

obtain the amplitudes of dunes under flows with suspended load.

Ultimately, our linear analyses provide a possible explanation for the absence of dunes in turbidites: suspended load suppresses dune formation and facilitates plane-bed formation. Previous research has suggested that the formation of dunes is suppressed due to the insufficient time for dune development (Walker, 1965), the hysteresis effect under waning flow conditions (Endo and Masuda, 1997), the turbulence suppression by high suspended-sediment concentrations (Lowe, 1988), the lack

of a sharp near-bed density gradient (Arnott, 2012), and the effect of clay-sized sediment on bed rheology (Schindler et al., 2015). Although these interpretations could explain the absence of dune-scale cross-lamination in turbidites, we show that dune formation is suppressed without considering the above conditions. Therefore, the above conditions are not required to suppress dune formation (Figs. 2b, d, 4b, d and 5b). Instead, we propose that the development of dune-scale bed waves under turbidity currents is restricted by the presence of suspended load.

**4.2 Effect of suspension on coarse sediment bed**

In the diagram with $\tilde{D} = 1.20$ mm, the data with sheet flows plotted much above the upper limit of Fr for the stable region (Fig. 3). Sheet flows consist of a shear layer of bed-load that moves under high shear stress (Shields number is larger than 0.5) (Gao, 2008). A few past experimental studies have observed that plane bed develops beneath sheet flows on coarse sediment beds in open-channel flows (Williams, 1970; Hernandez-Moreira et al., 2020). The difference in hydraulic properties between

standard bed-load and sheet flows could result in the disagreement between the stability diagrams and observational data. For example, the vertical velocity profile of an open-channel flow takes a logarithmic form (Keulegan, 1938), whereas that of sheet flows takes a power form (Sumer et al., 1996) or can be obtained by solving the differential equations (Egashira, 1997). In





addition, pressures of static interparticle contacts and inelastic particle collisions are not negligible in sheet flows (Egashira, 1997). Considering these differences in hydraulic conditions, the stability fields of perturbations are affected by sheet flows.

Further, linear analyses considering sheet flows can be extended to analyses of debris flows and turbidity currents that have collisional layers (Sohn, 1997; Lanzoni et al., 2017). These topics can be further explored in future works.

## 5 Conclusions

We investigated the influence of suspended load on the formation of plane beds under open-channel flows. The stability diagrams show that the stable region for finer sediments is wider in the diagram with suspension than that without suspension.

Further, the published data of plane beds with suspension coincide well with the stability diagrams where the suspension was considered. Our theoretical analysis found that suspended load promotes the formation of plane beds and suppresses the formation of dunes on the fine-grained bed. These results suggest that dune-scale cross lamination is absent in turbidites because the development of dunes in turbidity currents is restricted by the presence of suspended load. In addition, our analysis displays that the data pertaining to sheet flows deviate from the stable region. Additional theoretical work is required in order to examine

whether the plane bed under sheet flow can be interpreted as a stable condition or not.

*Code and data availability.* The datasets and codes used for this study can be found at [url to be updated at acceptance]. Unpublished data used for the analysis were cited from the dataset of Brownlie (2018).

## Appendix A: Linear analysis

In Sect. 2.2.1–2.2.6, we formulated the hydrodynamics, the sediment transport model, and the basic state. Here, we solve the
equations obtained in the above sections.

We impose an infinitesimal perturbation on the basic state. All the variables are modified using a small amplitude $A$ and a complex angular frequency of the perturbation $\omega$ as follows:

$$
\begin{aligned}
(\psi, p, h, Z, R, B, c) = {} & (\psi_0, p_0, 1, 0, R_0, B_0, c_0) \\
& + A(\psi_1, p_1, H_1, Z_1, R_1, B_1, c_1) \exp\left[\mathrm{i}\left(k\xi - \omega t\right)\right]
\end{aligned}
\tag{A1}
$$

The subscript 1 denotes a variable at $\mathcal{O}(A)$. By substituting Eq. (A1) into the governing equations and boundary conditions,
we can obtain the following equations at $\mathcal{O}(A)$:

$$
\mathcal{L}^{\psi}(\eta)\,\psi_1(\eta) + \mathcal{L}^{h}(\eta)\,H_1 + \mathcal{L}^{R}(\eta)\,R_1 = 0
\tag{A2}
$$

$$
\mathrm{i}k p_1(\eta) + \mathcal{P}^{\psi}(\eta)\,\psi_1(\eta) + \mathcal{P}^{h}(\eta)\,H_1 + \mathcal{P}^{R}(\eta)\,R_1 = 0
\tag{A3}
$$





Here, $\mathcal{L}^{\phi}$ and $\mathcal{P}^{\phi}$ $(\phi = \psi, h, R)$ are linear operators. The specific forms of $\mathcal{L}^{\phi}$ and $\mathcal{P}^{\phi}$ are skipped herein. With the use of the boundary conditions (Eqs. (35) and (36)), we get:

$$\psi_1(1) = 0 \tag{A4}$$

$$p_1(1) = 0 \tag{A5}$$

$$\psi_1(0) = 0 \tag{A6}$$

$$\left.\frac{\partial \psi_1}{\partial \eta}\right|_{\eta=0} = 0 \tag{A7}$$

Additionally, Eqs. (A3) and (A5) give:

$$\mathcal{P}^{\psi}(1)\,\psi_1(1) + \mathcal{P}^h(1)\,H_1 + \mathcal{P}^R(1)\,R_1 = 0 \tag{A8}$$

We employ a spectral collocation method using Chebyshev polynomials to solve the above differential equations. We expand $\psi_1$ using the Chebyshev polynomials as follows:

$$\psi_1 = \sum_{n=0}^{N} a_n T_n(\zeta) \tag{A9}$$

where $a_n$ is the coefficient for the $n$-th order Chebyshev polynomial $T_n$ and $\zeta$ is the independent variable of the Chebyshev
polynomials defined in the domain $[-1, 1]$. In this study, we transform $\zeta$ using the following equation to improve the calculation accuracy:

$$\zeta = 2\left\{\frac{\ln\left[(\eta + R_0)/R_0\right]}{\ln\left[(1 + R_0)/R_0\right]}\right\} - 1 \tag{A10}$$

The above functions are substituted into Eq. (A2); then, we evaluate the equation at the Gauss-Labatte points, which are defined as:

$$\zeta_j = \cos\left(\frac{j\pi}{N+2}\right) \quad , \quad j = 1, 2, ..., N+1 \tag{A11}$$

By combining the governing equations, boundary conditions, and closure assumptions, we obtain the following system of linear algebraic equations:

$$\mathbf{L}\mathbf{a} = \mathbf{M}R_1 \tag{A12}$$





with

$$\mathbf{L} = \begin{pmatrix} T_0(-1) & \cdots & T_N(-1) & 0 \\ \breve{\langle} T_0(-1) & \cdots & \breve{\langle} T_N(-1) & 0 \\ T_0(1) & \cdots & T_N(1) & 0 \\ \breve{\mathcal{P}}^\psi T_0(1) & \cdots & \breve{\mathcal{P}}^\psi T_N(1) & \breve{\mathcal{P}}^h \\ \breve{\mathcal{L}}^\psi T_0(\zeta_2) & \cdots & \breve{\mathcal{L}}^\psi T_N(\zeta_2) & \breve{\mathcal{L}}^h \\ \vdots & \ddots & \vdots & \vdots \\ \breve{\mathcal{L}}^\psi T_0(\zeta_{N-2}) & \cdots & \breve{\mathcal{L}}^\psi T_N(\zeta_{N-2}) & \breve{\mathcal{L}}^h \end{pmatrix} \tag{A13}$$

$$\mathbf{a} = (a_0, a_1, \ldots, a_N, D_1) \tag{A14}$$

$$\mathbf{M} = \left(0, 0, 0, \breve{\mathcal{P}}^R, \breve{\mathcal{L}}^h, \ldots, \breve{\mathcal{L}}^h\right) \tag{A15}$$

where a check mark ($\breve{\ }$) denotes a linear operator associated with variable transformation from $\eta$ to $\zeta$. We obtain the following solution from Eq. (A12):

$$\mathbf{a} = \mathbf{L}^{-1} \mathbf{M} R_1 \tag{A16}$$

Additionally, Eqs. (A9) and (A16) give:

$$\psi_1 = \psi_1^*(\eta) R_1 \tag{A17}$$

$$H_1 = H_1^* R_1 \tag{A18}$$

Similarly, we solve the eigenvalue problems for the sediment transport equations. By substituting Eq. (A1) into Eq. (24), we obtain the following equations at the order of $\mathcal{O}(A)$:

$$\mathcal{C}^c c_1(\eta) + \mathcal{C}^\psi(\eta) \psi_1(\eta) + \mathcal{C}^H H_1 + \mathcal{C}^R R_1 = 0 \tag{A19}$$

Based on Eqs. (A17) and (A18), we obtain:

$$\mathcal{C}^c c_1(\eta) + \left(\mathcal{C}^\psi(\eta) \psi_1^*(\eta) + \mathcal{C}^H H_1^* + \mathcal{C}^R\right) R_1 = 0 \tag{A20}$$

The boundary conditions give:

$$\mathcal{S}^c c_1(1) + \left(\mathcal{S}^\psi(1) \psi_1^*(1) + \mathcal{S}^H H_1^* + \mathcal{S}^R\right) R_1 = 0 \tag{A21}$$

$$\mathcal{B}^c c_1(0) + \left(\mathcal{B}^\psi(0) \psi_1^*(0) + \mathcal{B}^H H_1^* + \mathcal{B}^R\right) R_1 = 0 \tag{A22}$$

Here, $\mathcal{C}^\phi$, $\mathcal{S}^\phi$ and $\mathcal{B}^\phi$ ($\phi = \psi, h, R, c$) are the linear operators.

We expand $c_1$ using Chebyshev polynomials as follows:

$$c_1 = \sum_{n=0}^{N} b_n T_n(\zeta) \tag{A23}$$





The system is evaluated at the Gauss-Labatte points, then we obtain:

$$\mathbf{K}\mathbf{b} = \mathbf{N}R_1 \tag{A24}$$

with

$$\mathbf{K} = \begin{pmatrix} \check{\mathcal{B}}^c T_0(-1) & \cdots & \check{\mathcal{B}}^c T_N(-1) \\ \check{\mathcal{S}}^c T_0(1) & \cdots & \check{\mathcal{S}}^c T_N(-1) \\ \check{\mathcal{C}}^c T_0(\zeta_1) & \cdots & \check{\mathcal{C}}^c T_N(\zeta_1) \\ \vdots & \ddots & \vdots \\ \check{\mathcal{C}}^c T_0(\zeta_{N-1}) & \cdots & \check{\mathcal{C}}^c T_N(\zeta_{N-1}) \end{pmatrix} \tag{A25}$$

$$\mathbf{b} = (b_0, b_1, \ldots, b_N) \tag{A26}$$

$$\mathbf{N} = - \begin{pmatrix} \check{\mathcal{B}}^\psi \psi_1^*(-1) + \check{\mathcal{B}}^h H_1^* + \check{\mathcal{B}}^R \\ \check{\mathcal{S}}^\psi \psi_1^*(1) + \check{\mathcal{S}}^h H_1^* + \check{\mathcal{S}}^R \\ \check{\mathcal{C}}^\psi \psi_1^*(\zeta_1) + \check{\mathcal{C}}^h H_1^* + \check{\mathcal{C}}^R \\ \vdots \\ \check{\mathcal{C}}^\psi \psi_1^*(\zeta_{N-1}) + \check{\mathcal{C}}^h H_1^* + \check{\mathcal{C}}^R \end{pmatrix} \tag{A27}$$

The coefficient $b_n$ is derived as:

$$\mathbf{b} = \mathbf{K}^{-1}\mathbf{N}R_1 \tag{A28}$$

Therefore, the following equation is obtained:

$$c_1(\eta) = c_1^*(\eta)R_1 \tag{A29}$$

By substituting Eqs. (A17), (A18), and (A29) into Exner's equation (Eq. (61)), the complex angular frequency $\omega$ is obtained in the following form:

$$\omega = \omega(k, \mathrm{Fr}, C_\mathrm{z}, \mathrm{Re_p}) = \omega_\mathrm{r} + \mathrm{i}\omega_\mathrm{i} \tag{A30}$$

where $\omega_\mathrm{i}$ corresponds to the growth rate of the perturbation.

Here, using $\mathrm{Re_p} = \mathrm{Re_p}(D) = \mathrm{Re_p}(\tilde{D}, \tilde{h}_0)$ (Eq. (27)) and $C_\mathrm{z} = C_\mathrm{z}(R_0) = C_\mathrm{z}(\tilde{D}, \tilde{h}_0)$ (Eq. (52)), we can rewrite Eq. (A30)

as:

$$\omega = \omega\left(k, \mathrm{Fr}, \tilde{D}, \tilde{h}_0\right) \tag{A31}$$

Thus, we can obtain the growth rate $\omega_\mathrm{i}$ as a function of $k$ for a given combination of $\left(\mathrm{Fr}, \tilde{D}, \tilde{h}_0\right)$.





*Author contributions.* KO and NI performed the linear stability analysis. HN and NI contributed to the interpretation of the results. KO wrote the manuscript and prepared the figures, and then HN and NI provided feedback on the manuscript and figures.

*Competing interests.* The authors declare no competing interests.

*Acknowledgements.* This work was supported by the Japan Society for the Promotion of Science (JSPS) Grant-in-Aid (KAKENHI) Grant Number 18J22211. We would like to express our gratitude to Robert Dorrell for his comments. We are thankful to anonymous referees for their insightful comments on earlier versions of the manuscript.



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

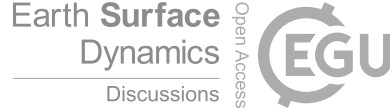

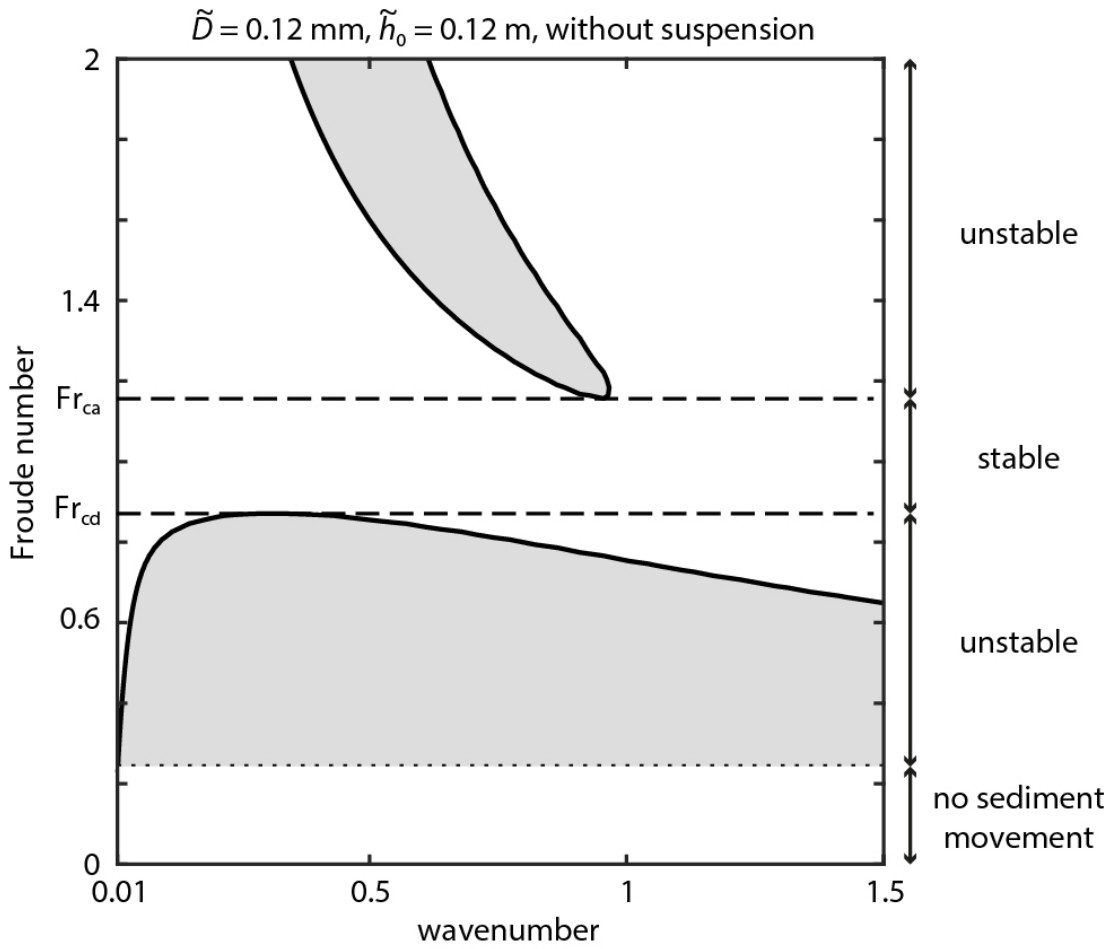

**Figure 1.** Contour map of perturbation growth rate $\omega_i$ without suspension. Sediment diameter and flow depth were set to $\tilde{D} = 0.12$ mm and $\tilde{h}_0 = 0.1201$ m, respectively. The dotted line denotes the threshold of sediment motion. The dashed lines denote the critical Froude numbers $\mathrm{Fr}_{cd}$ and $\mathrm{Fr}_{ca}$ for instabilities. The region where the growth rate is positive is highlighted in grey.





**Figure 2.** Contour maps of the dominant wavenumbers of perturbations with a fixed sediment diameter $\tilde{D}$. Symbols are observational data. a, $\tilde{D} = 0.12$ mm without suspension. b, $\tilde{D} = 0.12$ mm with suspension. c, $\tilde{D} = 0.25$ mm without suspension. d, $\tilde{D} = 0.25$ mm with suspension. a and b, The range of $\tilde{D}$ of observational data is from 0.0883 mm to 0.163 mm. c and d, The range of $\tilde{D}$ of observational data is from 0.184 mm to 0.34 mm.

Earth **Surface**
Dynamics
Discussions

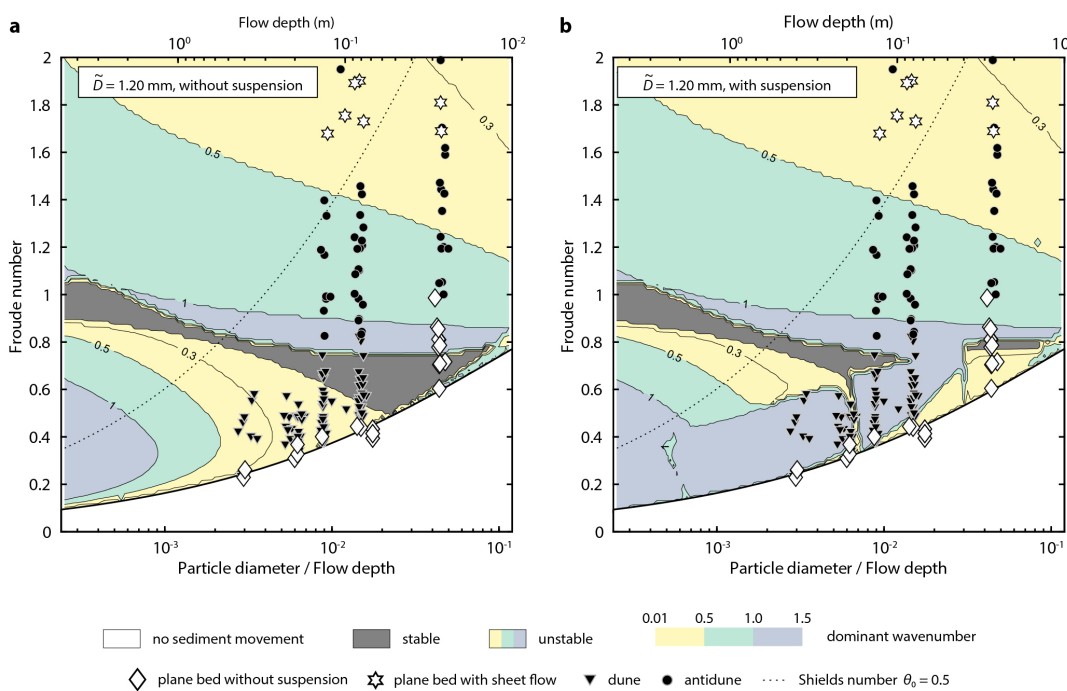

**Figure 3.** Contour maps of dominant wavenumbers of perturbations. Symbols are observational data. The sediment diameter $\tilde{D}$ was set to 1.20 mm. a, Without suspension. b, With suspension. The range of $\tilde{D}$ of observational data is from 0.883 mm to 1.63 mm.







**Figure 4.** Contour maps of the dominant wavenumbers of perturbations with a fixed flow depth $\tilde{H}$. Symbols are observational data. a, $\tilde{H} = 0.15$ m without suspension. b, $\tilde{H} = 0.15$ m with suspension. c, $\tilde{H} = 0.30$ m without suspension. d, $\tilde{H} = 0.30$ m with suspension. a and b, The range of $\tilde{H}$ of observational data is from 0.11 m to 0.21 m. c and d, The range of $\tilde{H}$ of observational data is from 0.21 m to 0.42 m.





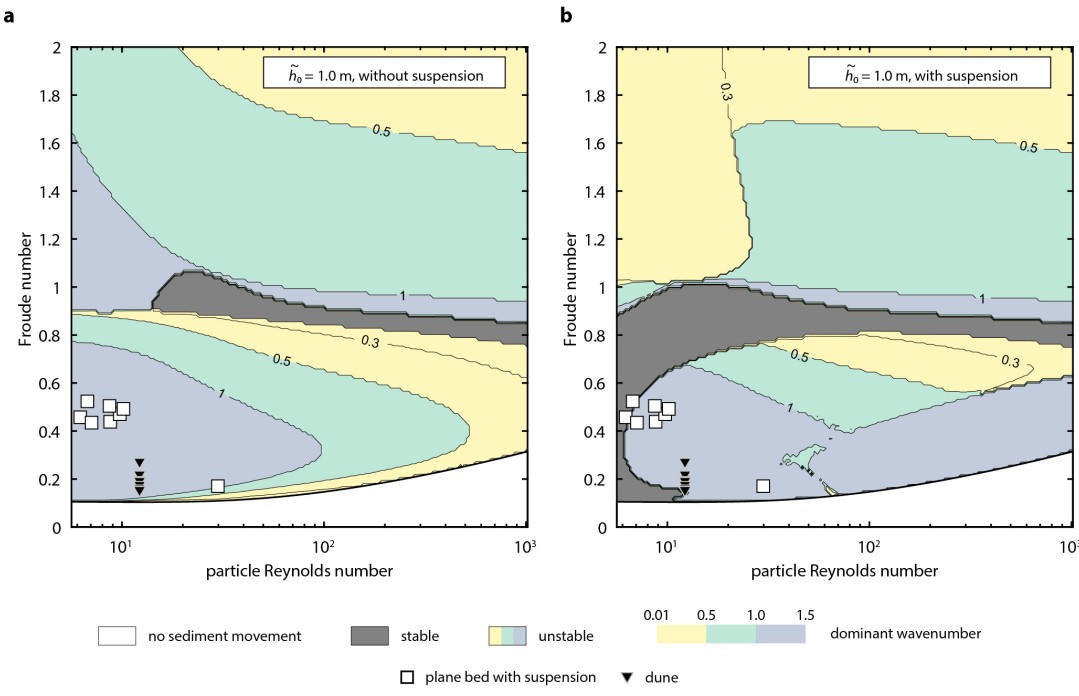

**Figure 5.** Contour maps of the dominant wavenumbers of perturbations with a fixed flow depth $\tilde{H}$. Symbols are observational data. a, $\tilde{H} = 1.0$ m without suspension. b, $\tilde{H} = 1.0$ m with suspension. The range of $\tilde{H}$ of observational data is from 0.7 m to 1.41 m.





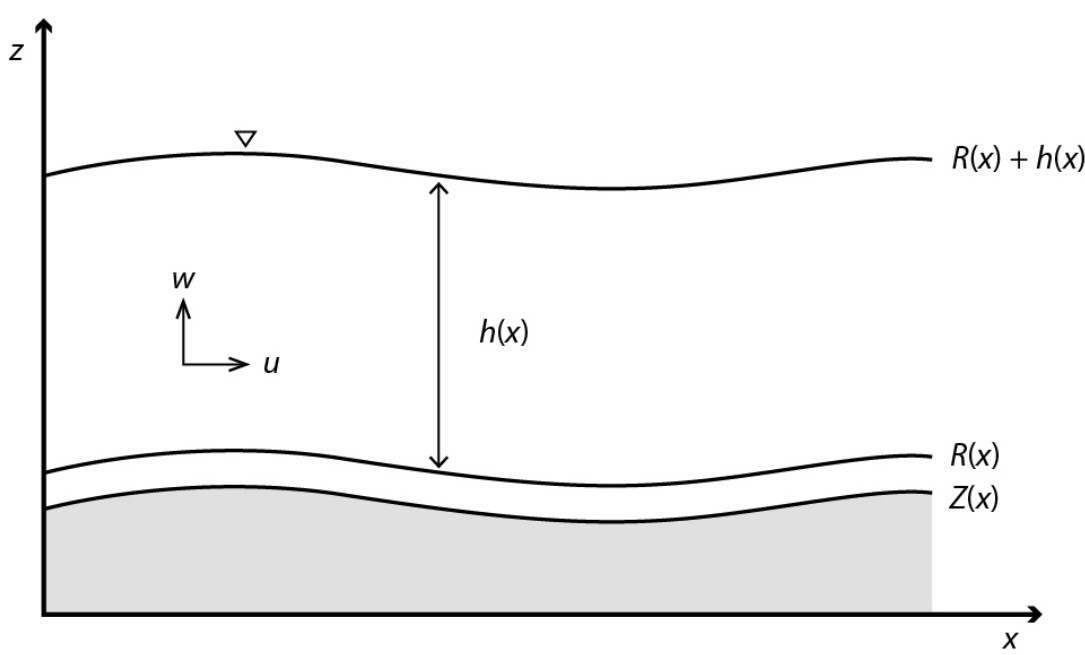

**Figure A1.** Conceptual diagram of the flow. The dimensionless parameters $u$ and $w$ are the flow velocities in $x$- and $z$- directions, respectively, $h$ is the flow depth, $Z$ denotes the bed height, and $R$ is the height of the reference level at which the flow velocity is assumed to vanish in a logarithmic law.





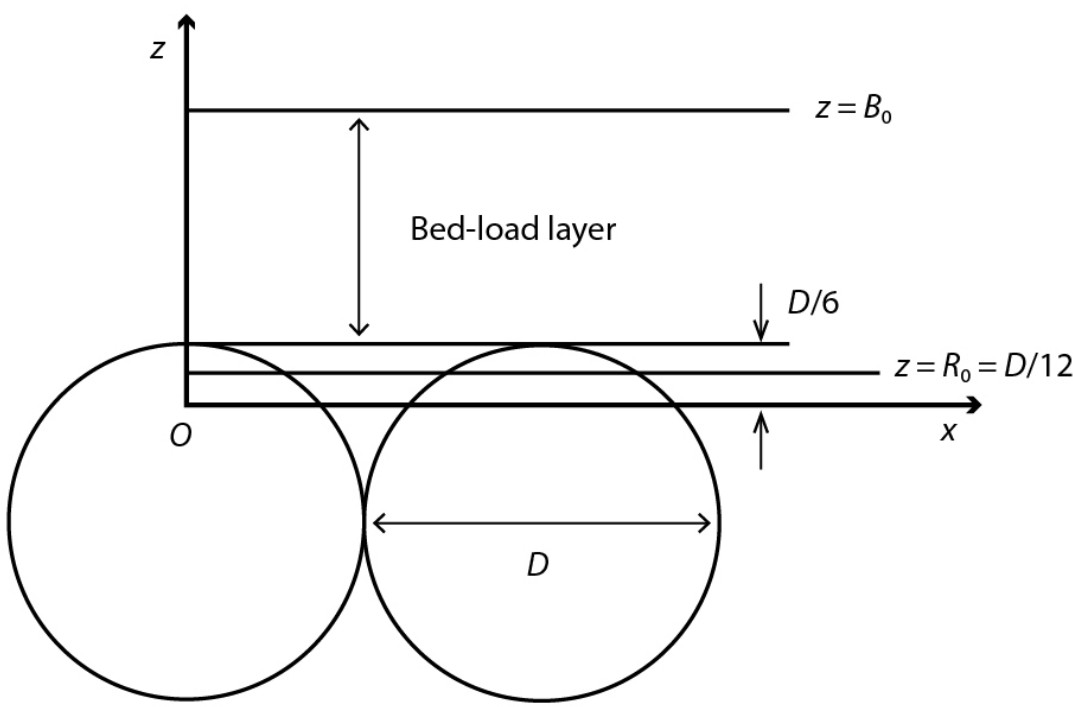

**Figure A2.** Conceptual diagram of the sediment bed. The origin of $z$-direction is denoted by $O$. The parameter $D$ is the dimensionless diameter of a bed particle, $B_0$ is the height of the top of the bed-load layer in the basic state, and $R_0$ is the height of reference level in the basic state.



**Table A1.** Summary of data used for the stability diagram with $\tilde{D} = 0.12$ mm.

| Reference | # of points | flow depth $\tilde{h}$ [m] | flow velocity $\tilde{U}$ [m/s] | particle diameter $\tilde{D}$ [mm] | Froude number Fr | Source |
|---|---|---|---|---|---|---|
| **Plane bed with suspension** | | | | | | |
| Taylor (1971) | 2 | 0.0783 | 0.585 | 0.138 | 0.668 | Flume |
| Simons (1957) | 1 | 1.83 | 0.585 | 0.096 | 0.138 | Field |
| Culbertson et al. (1972) | 8 | 0.494–0.957 | 1.06–1.42 | 0.16–0.2 | 0.415–0.524 | Field |
| **Dunes** | | | | | | |
| Baird (2010) | 1 | 2.24 | 0.744 | 0.16 | 0.159 | Field |
| Shen et al. (1978) | 2 | 2.94–3.07 | 1.55–1.61 | 0.208–0.218 | 0.288–0.294 | Field |
| **Antidunes** | | | | | | |
| Tanaka (1970) | 5 | 0.0443–0.11 | 0.658–1.14 | 0.145 | 0.852–1.38 | Flume |





Earth **Surface** Dynamics Discussions Open Access EGU

**Table A2.** Summary of data used for the stability diagram with $\tilde{D} = 0.25$ mm.

| Reference | # of points | flow depth $\tilde{h}$ [m] | flow velocity $\tilde{U}$ [m/s] | particle diameter $\tilde{D}$ [mm] | Froude number Fr | Source |
|---|---|---|---|---|---|---|
| **Plane bed without suspension** | | | | | | |
| Taylor (1971) | 7 | 0.0606–0.061 | 0.198–0.229 | 0.215–0.248 | 0.256–0.296 | Flume |
| **Plane bed with suspension** | | | | | | |
| Bridge and Best (1988) | 2 | 0.1 | 0.9–0.98 | 0.3 | 0.909–0.990 | Flume |
| Guy et al. (1966) | 10 | 0.155–0.241 | 0.881–1.29 | 0.19–0.33 | 0.686–1.05 | Flume |
| Taylor (1971) | 4 | 0.0788–0.114 | 0.692–0.878 | 0.228 | 0.778–0.838 | Flume |
| Culbertson et al. (1972) | 3 | 0.284–0.969 | 0.457–1.52 | 0.2–0.24 | 0.264–0.492 | Field |
| **Dunes** | | | | | | |
| Bridge and Best (1988) | 2 | 0.1 | 0.6–0.8 | 0.3 | 0.606–0.808 | Flume |
| Gee (1975) | 6 | 0.0454–0.105 | 0.305–0.920 | 0.305 | 0.325–1.04 | Flume |
| Guy et al. (1966) | 31 | 0.140–0.344 | 0.421–0.820 | 0.19–0.33 | 0.318–0.628 | Flume |
| Naqshband et al. (2014) | 2 | 0.25 | 0.64–0.8 | 0.29 | 0.409–0.511 | Flume |
| Abdel-Fattah et al. (2004), Wilbers (2004) | 6 | 4.03–5.72 | 0.31–0.75 | 0.239–0.322 | 0.0493–0.118 | Field |
| Baird (2010) | 2 | 1.67–2.33 | 0.597–1.33 | 0.21 | 0.148–0.278 | Field |
| Gabel (1993) | 4 | 0.4–0.43 | 0.61–0.65 | 0.31–0.33 | 0.301–0.320 | Field |
| Julien (1992) | 28 | 6.6–19.5 | 1.3–1.55 | 0.2–0.33 | 0.094–0.186 | Field |
| Mezaki (1973) | 8 | 0.9–1.35 | 0.49–0.83 | 0.21 | 0.154–0.272 | Field |
| Neill (1969) | 1 | 3.05 | 1.10 | 0.34 | 0.201 | Field |
| Shen et al. (1978) | 16 | 2.78–4.94 | 1.37–1.73 | 0.193–0.266 | 0.240–0.309 | Field |
| **Antidunes** | | | | | | |
| Foley (1975) | 3 | 0.0305–0.0473 | 0.546–0.692 | 0.28 | 0.813–1.26 | Flume |
| Fukuoka et al. (1982) | 15 | 0.0209–0.0569 | 0.349–0.93 | 0.19 | 0.760–1.45 | Flume |
| Guy et al. (1966) | 13 | 0.0914–0.204 | 1.06–1.62 | 0.19–0.33 | 0.892–1.30 | Flume |
| Kennedy (1961) | 15 | 0.0448–0.106 | 0.637–1.05 | 0.233 | 0.798–1.49 | Flume |





**Table A3.** Summary of data used for the stability diagram with $\tilde{D} = 1.20$ mm.

| Reference | # of points | flow depth $\tilde{h}$ [m] | flow velocity $\tilde{U}$ [m/s] | particle diameter $\tilde{D}$ [mm] | Froude number Fr | Source |
|---|---|---|---|---|---|---|
| **Plane bed without suspension** | | | | | | |
| Guy et al. (1966) | 6 | 0.149–0.314 | 0.381–0.454 | 0.93 | 0.229–0.368 | Flume |
| Taylor (1971) | 6 | 0.061 | 0.305–0.335 | 1.07 | 0.394–0.433 | Flume |
| Williams (1970) | 14 | 0.0283–0.155 | 0.332–0.558 | 1.35 | 0.401–0.986 | Flume |
| **Plane bed with sheet flow** | | | | | | |
| Hernandez-Moreira et al. (2020) | 2 | 0.091–0.1 | 1.58–1.66 | 0.85–1.09 | 1.59–1.75 | Flume |
| Williams (1970) | 6 | 0.0299–0.143 | 0.914–1.99 | 1.35 | 1.68–1.90 | Flume |
| **Dunes** | | | | | | |
| Blom et al. (2003) | 1 | 0.245 | 0.69 | 1.3 | 0.445 | Flume |
| Gee (1975) | 4 | 0.0622–0.146 | 0.449–0.584 | 1 | 0.488–0.575 | Flume |
| Guy et al. (1966) | 14 | 0.140–0.338 | 0.488–0.951 | 0.93 | 0.370–0.583 | Flume |
| Williams (1970) | 71 | 0.0872–0.223 | 0.448–0.917 | 1.35 | 0.343–0.825 | Flume |
| Shinohara and Tsubaki (1959) | 2 | 0.202–0.372 | 0.7–0.752 | 1.33 | 0.394–0.497 | Field |
| Sukhodolov et al. (2006) | 1 | 0.35 | 0.44 | 1 | 0.238 | Field |
| **Antidunes** | | | | | | |
| Fukuoka et al. (1982) | 1 | 0.0355 | 0.852 | 1.6 | 1.44 | Flume |
| Tanaka (1970) | 1 | 0.0807 | 1.74 | 0.91 | 1.95 | Flume |
| Williams (1970) | 43 | 0.0271–0.157 | 0.466–1.69 | 1.35 | 0.826–2 | Flume |





Earth **Surface** Dynamics
Discussions

**Table A4.** Summary of data used for the stability diagram with $\tilde{h} = 0.15$ m.

| Reference | # of points | flow depth $\tilde{h}$ [m] | flow velocity $\tilde{U}$ [m/s] | particle diameter $\tilde{D}$ [mm] | Froude number Fr | Source |
|---|---|---|---|---|---|---|
| **Plane bed without suspension** | | | | | | |
| Guy et al. (1966) | 7 | 0.131–0.195 | 0.265–1.31 | 0.19–0.93 | 0.234–0.947 | Flume |
| Williams (1970) | 1 | 0.155 | 0.494 | 1.35 | 0.401 | Flume |
| **Plane bed with suspension** | | | | | | |
| Guy et al. (1966) | 11 | 0.155–0.204 | 0.991–1.61 | 0.19–0.47 | 0.708–1.30 | Flume |
| Taylor (1971) | 2 | 0.112–0.114 | 0.866–0.878 | 0.228 | 0.819–0.838 | Flume |
| **Plane bed with sheet flow** | | | | | | |
| Williams (1970) | 1 | 0.143 | 1.99 | 0.135 | 1.68 | Flume |
| **Dunes** | | | | | | |
| Gee (1975) | 1 | 0.146 | 0.583 | 1 | 0.488 | Flume |
| Guala et al. (2014) | 2 | 0.193–0.209 | 0.661–0.717 | 1.8 | 0.48–0.5 | Flume |
| Guy et al. (1966) | 43 | 0.125–0.210 | 0.396–0.783 | 0.19–0.93 | 0.306–0.628 | Flume |
| Leclair and Bridge (2001) | 15 | 0.15–0.21 | 0.5–0.84 | 0.43–0.81 | 0.348–0.618 | Flume |
| Venditti et al. (2005) | 3 | 0.152–0.153 | 0.454–0.501 | 0.5 | 0.371–0.41 | Flume |
| Wijbenga and Klaassen (1983) | 3 | 0.2–0.21 | 0.485–0.49 | 0.78 | 0.346–0.35 | Flume |
| Wren et al. (2007) | 8 | 0.192–0.208 | 0.66–0.68 | 0.53 | 0.469–0.495 | Flume |
| Shinohara and Tsubaki (1959) | 1 | 0.202 | 0.7 | 1.33 | 0.497 | Field |
| **Antidunes** | | | | | | |
| Guy et al. (1966) | 32 | 0.113–0.204 | 1.18–1.88 | 0.19–0.47 | 0.891–1.66 | Flume |
| Tanaka (1970) | 1 | 0.11 | 1.1 | 0.145 | 1.06 | Flume |
| Guala et al. (2014) | 9 | 0.138–0.157 | 1.00–1.69 | 1.35 | 0.826–1.40 | Flume |



**Table A5.** Summary of data used for the stability diagram with $\tilde{h} = 0.30$ m.

| Reference | # of points | flow depth $\tilde{h}$ [m] | flow velocity $\tilde{U}$ [m/s] | particle diameter $\tilde{D}$ [mm] | Froude number Fr | Source |
|---|---|---|---|---|---|---|
| **Plane bed without suspension** | | | | | | |
| Guy et al. (1966) | 3 | 0.308–0.314 | 0.402–0.454 | 0.93 | 0.229–0.261 | Flume |
| **Plane bed with suspension** | | | | | | |
| Guy et al. (1966) | 4 | 0.213–0.241 | 1.05–1.16 | 0.19–0.54 | 0.686–0.787 | Flume |
| Culbertson et al. (1972) | 2 | 0.283–0.354 | 0.457–0.491 | 0.2–0.24 | 0.263–0.274 | Field |
| **Dunes** | | | | | | |
| Blom et al. (2003) | 1 | 0.245 | 0.69 | 1.3 | 0.445 | Flume |
| Guy et al. (1966) | 50 | 0.216–0.405 | 0.472–1.01 | 0.19–0.93 | 0.246–0.65 | Flume |
| Iseya (1984) | 3 | 0.231–0.395 | 0.584–0.933 | 0.57 | 0.388–0.474 | Flume |
| Kuhnle and Wren (2009) | 2 | 0.219–0.224 | 0.59–0.6 | 0.5–0.503 | 0.398–0.409 | Flume |
| Naqshband et al. (2014) | 2 | 0.25 | 0.64–0.8 | 0.29 | 0.409–0.511 | Flume |
| Wijbenga and Klaassen (1983) | 2 | 0.301 | 0.588 | 0.78 | 0.342 | Flume |
| Williams (1970) | 13 | 0.213–0.223 | 0.53–0.78 | 1.35 | 0.364–0.539 | Flume |
| Gabel (1993) | 5 | 0.34–0.42 | 0.61–0.65 | 0.31–0.39 | 0.301–0.345 | Field |
| Shinohara and Tsubaki (1959) | 1 | 0.372 | 0.752 | 1.33 | 0.394 | Field |
| Sukhodolov et al. (2006) | 1 | 0.35 | 0.44 | 1 | 0.237 | Field |



**Table A6.** Summary of data used for the stability diagram with $\tilde{h} = 1.0$ m.

| Reference | # of points | flow depth $\tilde{h}$ [m] | flow velocity $\tilde{U}$ [m/s] | particle diameter $\tilde{D}$ [mm] | Froude number Fr | Source |
|---|---|---|---|---|---|---|
| **Plane bed with suspension** | | | | | | |
| Simons (1957) | 1 | 1.32 | 0.612 | 0.349 | 0.17 | Field |
| Culbertson et al. (1972) | 7 | 0.747–1.11 | 1.19–1.66 | 0.19–0.21 | 0.435–0.524 | Field |
| **Dunes** | | | | | | |
| Mezaki (1973) | 8 | 0.9–1.35 | 0.49–0.83 | 0.21 | 0.154–0.272 | Field |