# Peer review of "Linear stability analysis of plane beds under flows with suspended load"

_Earth Surface Dynamics, 2022_

## Author Comment (AC1)

**Response to Reviewer 1's comments**

*General comments*
*The manuscript deals with a linear stability analysis of flow over an erodible bed with suspended load. The topic of dune-antidune formation has been deeply investigated in the past in terms of linear stability analysis, but the effect of suspended load has been neglected in recent theories, which assume bedload only. The inclusion of suspension represents therefore an interesting development. My main concern remains the same. There is no such thing as the "formation of a plane bed". Plane bed is not a bedform with an extremely small wavenumber (or an infinite wavelength) as mentioned at line 61 of the revised manuscript. Plane bed is the result of the absence of bedforms, which corresponds to the stable "upper plane bed" region where neither dunes nor antidunes form and the growth rate is negative. Indeed, the problem under investigation is the stability of a uniform flow over an erodible plane bed with active sediment transport. The focus of the paper must be on the effect of suspension on the formation of dunes and antidunes. If the unstable regions expand, the stable "upper plane bed" region shrinks, and viceversa. The choice of the governing parameters is unfortunate, with an unnecessary and awful mix of dimensional and non dimensional quantities which makes the analysis of the results quite cumbersome. Finally the discussion is too concise and leaves many points unaddressed. This is true for the conclusions as well.*

Thank you very much. We really appreciate your comments on our manuscript. Although the flat conditions of the bed can be interpreted as the state where the relatively large-scale bed waves are absent, such state of the bed is also classified as bedforms (e.g., Simons et al., 1965). Also, it is known that plane bed may low-relief bed waves at the hydraulic conditions (Bridge and Demico, 2008). To your specific comments, we have responded below.

*Specific comments*
*1 Please show stability plots in the Fr-k space. The Froude number is THE stability parameter for dune-antidune stability the sub-super critical character of these bedforms being well established. If you want to show the effect of suspension on the dune-antidune stability, you should start from the marginal curves (the boundaries of the unstable regions) in this space.*

Thank you for the comment. In order to compare the theoretical results with the observational data of plane beds, a single Fr-$k$ diagram is insufficient. Firstly, this study employed fully dimensionless parametric space for considering the stability of bedforms, as described in Line 72. Fr number, dimensionless flow depth ($D/H$), and the particle Reynolds number ($Re_p = \frac{\sqrt{RgDD}}{\nu}$) are used in the stability diagrams (Figs. 2–5). Observational data of bedforms were plotted in this dimensionless parametric space. Regarding the linear stability analysis results, as you pointed out, the plane bed is expressed as the condition where the growth rates of perturbation of the bed are negative for all wave numbers. Thus, a single Fr-$k$ diagram indicates a range of Fr number for producing plane bed (or dunes/anti-dunes) at the fixed values of the particle Reynolds number and the dimensionless flow depth ($D/H$). Therefore, we repeated the calculation to obtain Fr-$k$ diagrams with different values of $D/H$ and $Re_p$ to illustrate the stability region of the plane bed in the dimensionless parametric space. To avoid including too many numbers of Fr-$k$ diagrams in this manuscript, we plot only the resultant phase diagram. Several Fr-$k$ diagrams exhibiting representative flow conditions will be added in the appendix of the revised version of this manuscript.

*2 In your rebut to the original submission, you stated "Actually, the Shields stress does not necessarily increase with Fr because the Froude number is normalized by the square root of the product of the flow thickness and the gravity acceleration.". This is a nonsense. Stability plots in the Fr-k space are obtained for a constant value of the grain size to depth ratio D (or of the friction coefficient Cz), so that the Shields stress of the base uniform flow is strictly proportional to the square of the Froude number.*

Thank you for the comment. The Froude number is indeed proportional to the Shields stress when only a single Fr-$k$ diagram is considered. However, this study considers the stability condition of bedforms in the dimensionless parametric space employing Fr, $D/H$, and $Re_p$. In this parametric space, the high Shields stress can be attained with the lower value of the Froude number when $D/H$ is small. Thus, the suspended load can occur even at the lower values of the Fr number if the flow depth is sufficiently large.

*3 Related to the previous point, I confirm my concern: the role of suspension should become increasingly important as Froude is increased, hence moving from dunes to the upper plane bed region to antidunes. I expect marginal curves in the Fr-k space to be deformed by the effect of suspension, the smaller the grain size to depth ratio D the lower the value of the Shields parameter at which the marginal curves for dunes and antidunes are affected. I would like to see clearly this effect before wandering in the Fr-D space, where any information on the wavenumber*

*is lost and the unstable regions for dunes and antidunes overlap (although they remain distinct in the Fr-k space) because the upper limit for dunes in terms of Froude number may be higher than the lower limit for antidunes, expecially for finer materials.*

Thank you for the comment. We will present several Fr-$k$ diagrams to exhibit two cases: the upper limit for dunes in terms of Froude number is higher or lower than the lower limit for antidunes.

*4 Stop using dimensional parameters! Use Rep instead of the fixed dimensional grain size in figures 2 and 3. Use the grain size to depth ratio instead of the fixed dimensional flow depth in figures 4 and 5.*

We incorporate your comment. We already employed the particle Reynolds number in Figures 2 and 3, whereas the figure legend remained to indicate the dimensional value. We initially considered that the dimensional values of the flow depth could be helpful to understand the result intuitively in Figures 4 and 5, but it will be changed to the $D/H$ values in the revised manuscript.

*5 The wavenumber of maximum amplification is difficult to read in regions where dunes and antidunes overlap. Please use the growth rate of maximum amplification instead and show the curves Frca(D) and Frcd(D) that bound the instability regions in Figures 2, 3, 4 and 5. Moreover, are the latter of any help for the reader in order to understand the results of your analysis? Five lines of text in the manuscript (280-284) do not justify two pages of figures.*

Thank you for the comment. We reconsider to use the growth rate of maximum amplification to describe the diagram and will present several Fr-$k$ diagrams.

*6 Before attempting a comparison with experimental data, provide the reader with some plots of your stability analysis to explain your results and choose more wisely the values of the parameter you fix: in figure 2b dunes disappear, meaning that suspension completely inhibits dune formation. This does not help much to understand what happens in between. Moreover, some pictures are really obscure: what are those color leaks in figure 3b at D=0.007 and D=0.03? For such a coarse bed material suspension should be irrelevant in the dune region, whereas the plot is remarkably different from figure 3a in that region. Hard to explain.*

Thank you for the comment. We revised the Fr-$k$ diagrams and found

that the numerical instability can be occurred in the linear stability analysis with the coarse grain diameter or low Froude number. Therefore, we reconsidered the initial conditions of linear stability analysis and recalculated without the condition where the calculation may be numerically unstable. As a result, we obtained the new phase diagrams and confirmed that the change of the initial conditions will not affect the discussion section. However, the phase diagram for $\mathrm{Re_p} = 167$ ($\tilde{D} = 1.2\mathrm{mm}$) will be removed in revised manuscript. Also, we will plot the diagram using the dimensionless parameters and will present several Fr-$k$ diagrams.

---

## Author Comment (AC2)

**Response to Reviewer 2's comments**

*In this manuscript, the authors use linear stability analysis to show that suspended sediment load could promote the stability of plane beds for open-channel flows with fine bed-material sediment. They propose that this mechanism could explain the observations of parallel laminations in turbidites, which typically lack dune-scale cross stratification. The authors also use observational data to test their hypothesis. Overall, the manuscript is reasonably well-written; however, the writing and presentation still needs a lot of work to clarify the results and avoid repetition. Importantly, I found that this manuscript needs significant amount of work to clarify several aspects of the analysis before being ready for publication.*

*1. The definition of a plane bed in terms of dominant wave number seems rather confusing to me. By definition, a plane bed is not a bed form that has a large wavelength. So, defining the plane bed this way and then using linear stability analysis to find parameter space that correspond to a small dominant wave number seems odd to me. At least, there is no justification given for why this should correspond to a strict definition of a plane bed. This is a major point as this assumption is the foundation for the entire manuscript.*

Thank you for the comment. In this paper, the plane bed is defined as the bed state where the growth rate of the bed perturbation is negative for all wave numbers. Thus, the plane bed is supposed to be a completely flat condition in theoretical analysis. We will revise the manuscript to clarify this.

*2. The limits on the parameter space explored here needs justification. For example, in lines 73-77, the authors describe the range of particle sizes and flow depths explored but also state that they set the grain size to 3 values and flow depths to 3 values. How is it that the data could not be recast into only dimensionless terms without the need for using a mix of dimensional and dimensionless variables?*

Thank you for the comment. We already employed the particle Reynolds number in Figures 2 and 3, whereas the figure legend remained to indicate the dimensional value. We initially considered that the dimensional values of the flow depth could be helpful to understand the result intuitively in Figures 4 and 5, but it will be changed to the $D/H$ values in the revised manuscript.

*3. The authors need to give more detail about the observational data that is used to support their hypothesis. How are data from a range of grain sizes and*

*flow depths collated to plot on stability diagrams with a single value of grain size, for example? What is the sensitivity of these stability diagrams to the parameters?*

Thank you for the comment. We show the range of the particle diameter of the observed data plotted in the diagram in Lines 253–254. Also, we will check the sensitivity of the diagram when we change the range of the $Re_p$.

*4. What is the criterion for the success of the model? It appears from the results that a majority of the observations plotting in the stable region of the contour maps is enough to state that the model works. There is no discussion of how many points do not plot in the stable region and what it means for the model veracity. I think the authors need to lay out the metrics they will use to test the success of the model and then discuss how the field and flume data compare with this test. Right now, the entire model testing part of the manuscript is weak and arbitrary.*

Thank you for the comment. We will show the error rate which denotes the ratio of the number of plane bed data plotted on the unstable region to the whole number of plane bed data.

*5. The figures need some more explanation. It is not clear to the reader where each of these data points should lie in terms of model expectations? For example, I would expect that if larger fraction of actual plane bed data lining up with the stable region in the contour plots would be a model success but I don't see a lot of observational data matching up with stable regions on the contour plots. If I am mistaken about my interpretation here, then the authors need to do a better job of explaining the metrics for success of their model.*

Thank you for the comments. As you stated, we interpreted that the model with suspension works because a majority of the plane bed data plotted in the stable region of the contour maps. We will show the error rate to state that more data plotted in the stable region for the case with suspension compared to the case for the model without suspension.

---

## Author Comment (AC3)

**Response to Reviewer 3's comments**

*This paper conducts a stability analysis to explicitly include sediment suspension in order to determine the role of suspension on the suppression of bed topography. Their analysis shows that the presence of suspended load is a controlling factor on upper plane bed stability with implications for the understanding of hydrodynamics of deposits such as those formed by turbidites and other high suspended sediment concentration flows. The model framework was well described and understandable. However I do agree with most comments posed by Reviewers 1 and 2, particularly the contextualization and success criterion questions brought up by reviewer 2 and the appropriateness of using the dominant wavenumber to define the formation of plane bed.*

Thank you for the comment. As we replied to Reviewers 1 and 2, we will consider to describe the diagram using the growth rate and to show the error rate.

*There are a few additional contextualization issues to clarify and put the work into the broader picture should those prior issues be adequately addressed. First, in the abstract, there are a number of sentences that seem repetitive, and the mechanistic component of the role that suspension plays is not described. Again in the introduction, the hypothesized role of suspension is not mechanistically discussed. In the discussion, there is brief allusion to the fact that this analysis demonstrates that turbulent suppression for example is not required, but I think the exact mechanism by which the presence of suspended load is not fully described in the written work.*

Thank you for the comment. We will describe the mechanistic component in the abstract and the introduction. Also, we will not state that the turbulent suppression is not required, but it may contribute to the deformation of dunes, and thus the model can be improved by the inclusion of such effect in the future studies.

*I should note that I don't necessarily agree with the dimensional arguments made by Reviewer 1 - while it is certainly common practice and more relevant in predictive modeling to non-dimensionalize, the framing of the arguments in this paper does not necessarily require it in my opinion.*

Thank you for the comment. As you commented, it is common practice to use dimensionless parameters. Therefore, it will be changed to the $D/H$ values in the revised manuscript.